# Putting computational models of immunity to the test—An invited challenge to predict *B.pertussis* vaccination responses

Pramod Shinde[1], Lisa Willemsen[1], Michael Anderson[2], Minori Aoki[1], Saonli Basu[2], Julie G. Burel[1], Peng Cheng[3], Souradipto Ghosh Dastidar[2], Aidan Dunleavy[4], Tal Einav[1,5], Jamie Forschmiedt[2], Slim Fourati[6], Javier Garcia[3], William Gibson[7], Jason A. Greenbaum[8], Leying Guan[9], Weikang Guan[3], Jeremy P. Gygi[10], Brendan Ha[8], Joe Hou[11], Jason Hsiao[3], Yunda Huang[11,12], Rick Jansen[13], Bhargob Kakoty[2], Zhiyu Kang[2], James J. Kobie[14], Mari Kojima[1], Anna Konstorum[15,16], Jiyeun Lee[1], Sloan A. Lewis[1], Aixin Li[17], Eric F. Lock[2], Jarjapu Mahita[1], Marcus Mendes[1], Hailong Meng[15], Aidan Neher[2], Somayeh Nili[1], Lars Rønn Olsen[18], Shelby Orfield[1], James A. Overton[19], Nidhi Pai[2], Cokie Parker[20], Brian Qian[3], Mikkel Rasmussen[1], Joaquin Reyna[21,22], Eve Richardson[1], Sandra Safo[2], Josey Sorenson[2], Aparna Srinivasan[2], Nicola Thrupp[1], Rashmi Tippalagama[1], Raphael Trevizani[1,23], Steffen Ventz[2], Jiuzhou Wang[2], Cheng-Chang Wu[2], Ferhat Ay[5,21,22], Barry Grant[3], Steven H. Kleinstein[10,15], Bjoern Peters[1,5]*

1 Center for Vaccine Innovation, La Jolla Institute for Immunology, La Jolla, California, United States of America, 2 Division of Biostatistics and Health Data Science, University of Minnesota, Minneapolis, Minnesota, United States of America, 3 Department of Molecular Biology, School of Biological Sciences, University of California, San Diego, La Jolla, California, United States of America, 4 School of Statistics, University of Minnesota, Minneapolis, Minnesota, United States of America, 5 Department of Medicine, University of California San Diego, San Diego, California, United States of America, 6 Department of Medicine, Division of Allergy and Immunology, Feinberg School of Medicine and Center for Human Immunobiology, Northwestern University, Chicago, Illinois, United States of America, 7 Vaccine Research Center, National Institute of Allergy and Infectious Disease, National Institute of Health, Bethesda, Maryland, United States of America, 8 LJI Bioinformatics Core, La Jolla Institute for Immunology, La Jolla, California, United States of America, 9 Department of Biostatistics, Yale School of Public Health, New Haven, Connecticut, United States of America, 10 Program in Computational Biology & Bioinformatics, Yale University, New Haven, Connecticut, United States of America, 11 Vaccine and Infectious Disease Division, Fred Hutchinson Cancer Research Center, Seattle, Washington, United States of America, 12 Department of Global Health, University of Washington, Seattle, Washington, United States of America, 13 Biostatistics Core, Masonic Cancer Center, University of Minnesota, Minneapolis, Minnesota, United States of America, 14 Department of Medicine, Division of Infectious Diseases, University of Alabama at Birmingham, Birmingham, Alabama, United States of America, 15 Department of Pathology, Yale School of Medicine, New Haven, Connecticut, United States of America, 16 Laboratory for Systems Biology, University of Florida, Gainesville, Florida, United States of America, 17 Division of Epidemiology and Community Health, University of Minnesota, Minneapolis, Minnesota, United States of America, 18 Department of Immunology and Microbiology, LEO Foundation Skin Immunology Research Center, University of Copenhagen, Copenhagen, Denmark, 19 Knocean, Inc., Toronto, Canada, 20 National Institute of Allergy and Infectious Diseases, National Institute of Health, Bethesda, Maryland, United States of America, 21 Center for Autoimmunity and Inflammation, La Jolla Institute for Immunology, La Jolla, California, United States of America, 22 Bioinformatics and Systems Biology Graduate Program, University of California, San Diego, California, United States of America, 23 Fundação Oswaldo Cruz, Fiocruz - Ceará, Brazil

* bpeters@lji.org

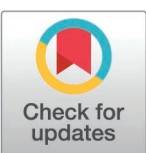

## Abstract

Systems vaccinology studies have been used to build computational models that predict individual vaccine responses and identify the factors contributing to differences in outcome. Comparing such models is challenging due to variability in study designs. To

**Data availability statement:** The training and challenge data for this prediction challenge can be found at https://doi.org/10.5281/zenodo.14968772 and at https://www.cmi-pb.org/downloads/cmipb_challenge_datasets/legacy/2nd_challenge/2024-02-02/2nd_challenge/. All methods developed by contestants are deposited on GitHub at https://github.com/topics/2nd-cmipb-challenge.

**Funding:** Research reported in this publication was supported by the National Institute of Allergy and Infectious Diseases of NIH under award nos. U01AI150753 (BP), U01AI141995(BP), U19AI142742(BP) and U01AI187062 (BP). The funders had no role in study design, data collection and analysis, decision to publish, or preparation of the manuscript.

**Competing interests:** The authors have declared that no competing interests exist.

address this, we established a community resource to compare models predicting *B. pertussis* booster responses and generate experimental data for the explicit purpose of model evaluation. We here describe our second computational prediction challenge using this resource, where we benchmarked 49 algorithms from 53 scientists. We found that the most successful models stood out in their handling of nonlinearities, reducing large feature sets to representative subsets, and advanced data preprocessing. In contrast, we found that models adopted from literature that were developed to predict vaccine antibody responses in other settings performed poorly, reinforcing the need for purpose-built models. Overall, this demonstrates the value of purpose-generated datasets for rigorous and open model evaluations to identify features that improve the reliability and applicability of computational models in vaccine response prediction.

## Author summary

Systems vaccinology approaches have been crucial in developing computational models that not only predict individual vaccine responses but also elucidate the underlying factors driving these differences. In our study, we invited scientists to participate in a community-wide computational challenge focused on predicting *B. pertussis* booster responses. We received 49 models developed by 53 scientists and evaluated them using purpose-generated experimental data. Our analysis revealed that the most successful models excelled in managing complex, nonlinear data, efficiently reducing large feature sets to key predictive subsets, and employing advanced data preprocessing techniques. In contrast, models adopted from other vaccine response studies performed poorly, underscoring the need for approaches tailored specifically to the context of pertussis booster responses. By leveraging a community resource dedicated to model evaluation, our work demonstrates the critical importance of purpose-built datasets in strengthening rigorous and open comparisons of computational approaches. Overall, this study provides valuable insights into the features and methods that enhance the reliability and applicability of vaccine response predictions, advancing the field of systems vaccinology and moving us closer to personalized vaccine strategies.

## Introduction

Systems vaccinology aims to translate complex immunological data into actionable insights that can guide vaccination strategies. Achieving this requires integrating diverse datasets including genomic, proteomic, and transcriptomic data, to evaluate the systemic response to vaccination and build computational models of the vaccine-induced immune responses [1–3]. As a scientific community, we are advancing towards this goal by expanding cohort sizes, establishing meta-analyses involving a broad range of immune responses, and continuously integrating diverse datasets from single vaccines [4–6] as well as multiple vaccines [7,8] together. These efforts aim to capture the full complexity of the immune system and enhance our understanding of vaccine efficacy and safety across different populations [4,8].

A key challenge in this endeavor is to objectively test the generalizability and reproducibility of the findings generated by models developed in different studies. It is well known for genome-wide association studies [9] that a given study can overemphasize dataset-specific results that do not replicate in other studies. The solution to this is to test previous findings

in independent future studies. This can be challenging for systems vaccinology as there is significant variability between studies in terms of their design, specimen collection timing, and assays used to evaluate results. In addition, systems vaccinology studies are resource intensive, reducing the incentive for generating validation datasets. This means that most systems vaccinology-based models are generated based on datasets analyzed at the point of their publication, but they are not tested further on independent data.

To address this challenge, we initiated CMI-PB (Computational Models of Immunity to Pertussis Booster; https://www.cmi-pb.org). Our main goal is to test computational models that predict the outcome of booster vaccination which is performed through a series of data releases and associated community prediction contests. We have previously completed the first of three planned contests (Table 1) - a 'dry-run' involving CMI-PB consortium members forming teams using different models to answer the contest questions [10]. In the current study, we report our findings on the second 'invited' contest that included a select group of scientists from the broader community who have previously published in systems vaccinology. The datasets from a total of 96 subjects (Table 1) as part of the first challenge [10] were made available as a training dataset to develop predictive models and we recruited a new cohort of 21 subjects, which was available as an unseen challenge dataset. We assessed over 49 computational models that applied various methodologies including classification-based techniques, such as Naive Bayes and Random Forest, regression-based approaches like Elastic Net, and various other strategies encompassing multi-omics integration, gene signature analysis, and module scoring. We describe these approaches, as well as general trends arising from a meta-analysis of all submissions. The full dataset, along with methods and scoring functions, are freely provided to the research community, and available to benchmark future algorithms in the field. The third public challenge will be open to community participation until November 2024.

## Results

This results section covers two components: Sections 1-3 describe the experience of setting up and running the invited prediction contest. Sections 4-7 describe the specific models developed and discuss their performance on the prediction tasks.

## 1. Invitation of a select group of challenge participants

Our goal for this 'invited challenge' was to recruit external participants while also keeping the number at a manageable level of < 50 teams to ensure we could provide individualized support. To identify potential participants, we first consulted CMI-PB investigators to identify

**Table 1. Overview of past and future CMI-PB prediction challenges.**

| Prediction challenge title | Contestants | Subjects in dataset | | Status |
|---|---|---|---|---|
| | | **Training** | **Challenge** | |
| **First:** Internal dry run | CMI-PB consortium | 60 (28 aP + 32 wP) | 36 (19 aP + 17 wP) | Concluded in May 2022 |
| **Second:** Invited challenge | Invited contestants | 96 (47 aP + 49 wP) | 21 (11 aP + 10 wP) | Concluded in January 2024 |
| **Third:** Open Challenge | Public | 117 (58 aP + 59 wP) | 54 (27 aP + 27 wP) | Announced in August 2024 |

Our commitment involves conducting three annual challenges. The first challenge was completed in May 2022 with participation from the CMI-PB consortium. The second challenge concluded in January 2024 and featured the CMI-PB consortium along with a limited number of invited contestants from outside the consortium. We will involve members of the public in the third challenge. The second challenge included training data from the first challenge and newly generated challenge data. Similarly, we will use the training and challenge data from previous challenges as the training data for future challenges and generate new data for testing purposes.

researchers with a proven track record in handling high-dimensional data and applying advanced modeling techniques. With that list in hand, we searched PubMed and Google Scholar for additional potential participants that authored papers with relevant keywords and/ or that were cited by publications from identified candidates. We then extended personalized invitations to them to participate in the CMI-PB Challenge. Initially, 10 out of the 50 invited participants confirmed that they or their lab members would be interested, while others mentioned conflicting schedules or time constraints as reasons for their inability to participate. Eventually, a total of 10 teams were formed, with three teams of 5-6 each from the University of Minnesota (including faculty, PhD and masters students), one team of 3 researchers from different institutions, and the six teams remaining consisting of individual researchers for a total of 27 external participants. In addition to the invitations sent to external participants, we also invited participants from the labs of CMI-PB investigators who were not directly involved with the project, resulting in 5 participating teams, plus 1 team consisting of 5 master students from University of California San Diego. Additionally, 2 teams from members of the CMI-PB Consortium participated in the challenge. In total, we gathered 18 participating teams for a total of 25 models submitted in the Challenge, which was a total of 53 people who participated in this invited challenge.

## 2. Summary of data sets and challenge tasks

**Providing experimental data for training and testing prediction models.** We generated data derived from more than 600 blood specimens collected from 117 subjects participating in a longitudinal study of *B. pertussis* booster vaccination. Blood specimens were collected on up to three days prior (day -30, -14, 0) and four days post-booster vaccination (day 1, 3, 7, and 14). For each specimen, we performed i) gene expression analysis (RNAseq) of bulk peripheral blood mononuclear cells (PBMC), ii) plasma cytokine concentration analysis, iii) cell frequency analysis of PBMC subsets, and iv) analysis of plasma antibodies against TdaP antigens (Fig 1; See Methods section for a detailed description of the profiling datasets). The repeat pre-vaccination samples were intended to give a stable estimate of baseline and variability. The contestants were supplied with pre- and post-vaccination data as a training dataset to build their prediction models that consisted of two independent cohorts, the 2020 and 2021 cohorts, for a total of 96 subjects, which are discussed in detail in two previous publications [10,11]. For this challenge, we generated data from 21 new subjects. Baseline (pre-vaccination) challenge data was made available to contestants. The post-vaccine response challenge data was hidden from the contestants and used as ground truth for model evaluation.

**Our data processing and harmonization approach.** As the training dataset includes two multi-omics datasets from the 2020 and 2021 cohorts, which involved changes in the researchers performing the assays, and in the manufacturer's setup of the assays, we are expecting batch effects that should be corrected before integrating them (Fig A in S1 Text). While data processing and normalization methods are inherently user-specific, the CMI-PB team has developed a standardized data processing approach inspired by the methodology used in the internal CMI-PB challenge [10]. This involves 1) identifying common features, 2) baseline median normalization, and 3) batch-effect correction.

As a first step, we identified what features should be included in our analysis. Features are analytes measured in individual omics assays, such as IL-6 in the plasma cytokine concentrations assay. After the removal of features that were not found in all datasets, we were left with 58,302 overlapping features (Fig 1A). Many of these features had low information content, especially for the transcriptomic assay. To address this, for gene expression, we filtered zero variance and mitochondrial genes and removed lowly expressed genes (genes with transcript per million [TPM] <1 in at least 30% of specimens). Similarly, we filtered features with zero

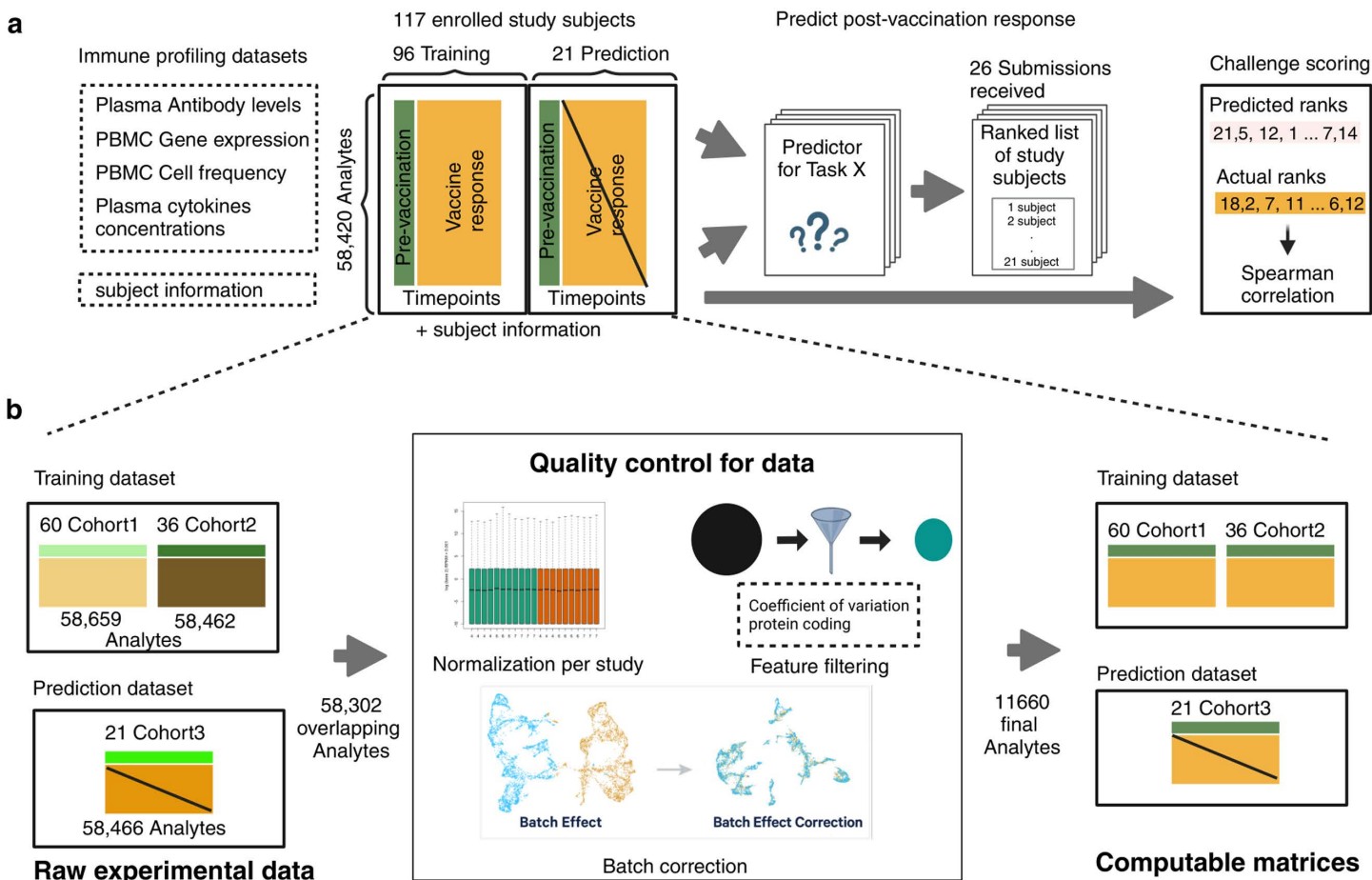

**Fig 1. Generation of multi-omics datasets for 117 study participants.** Contestants were provided with training datasets containing two cohorts (datasets 2020 and 2021), while the prediction dataset contained a newly generated cohort (dataset 2022). The training datasets contain pre-vaccination and post-vaccination immune response data, whereas the challenge dataset for 21 participants contains only pre-vaccination immune response data. Post-vaccination data will be released after the challenge ends and will be utilized to evaluate submitted models. Figure is created in https://BioRender.com.

variance from cytokine concentrations, cell frequency, and antibody assays. This resulted in 11,660 features, including 11,589 features from PBMC gene expression, 23 from PBMC cell frequency, 28 from plasma cytokine concentrations, and 20 from the plasma antibody measurements.

In the second step, we ran assay-specific data normalization. We performed baseline normalization on cell frequency, antibody titer, and cytokine concentration data. Specifically, we calculated the median using day zero time point data as a normalization factor per analyte and divided all values by this factor. We did not apply any normalization to the gene expression data as TPM conversion makes expression values comparable across samples. As a third step, we applied batch-effect correction on assay data within the training dataset to harmonize the data across 2020 and 2021 years. We employed the *ComBat* algorithm from the *sva* package, which adjusts for batch effects by modeling both batch and biological covariates [12,13]. After batch-effect correction, we validated the effectiveness of this step by examining the distribution of features across batches. We observed a significant reduction in cross-year batch-associated variability, confirming that the correction process was successful. This allowed us to move forward with a harmonized dataset for contestants for their analysis.

The challenge dataset underwent similar data processing and normalization to the training set to ensure consistency and comparability. This processed data, along with raw data, was made available in TSV files and R data object formats, and the codebase used to transform from raw to processed was made available through GitHub.

**Prediction tasks.** We formulated six tasks asking contestants to predict a ranking of subjects from the highest response to the lowest response for each task based only on the pre-vaccination immune state data (Table 2). In task 1.1, contestants were asked to predict plasma IgG levels against the pertussis toxin (PT) on day 14 post-booster vaccination. Task 1.2 consisted of predicting the fold change of the plasma IgG levels against PT between day 14 post-booster vaccination and baseline. Tasks 2.1 and 2.2 required contestants to predict the overall frequency of monocytes among PBMCs on day 1 post-booster vaccination and the corresponding fold change, respectively. Similarly, in tasks 3.1 and 3.2, the *CCL3* gene expression on day 3 post-booster vaccination and the corresponding fold change values compared to baseline needed to be predicted. This focus on 6 tasks that combine 3 targets with 2 readouts makes for a simpler setup compared to our previous competition.

Each team could enter submissions for up to 3 different models and was allowed to update their submissions until the deadline. In total, we received 25 submissions for this invited challenge from 20 participating teams. In addition, we constructed 2 control models and incorporated 22 models previously identified from the literature, bringing the total number of models evaluated to 49. All teams provided detailed information about their computational methods and deposited their source code on GitHub, as listed in S1 Note.

## 3. Establishing control models and literature models

We established two simple control models that set a baseline of what more complex models should outperform (Schematically shown in Fig B in S1 Text). Control model 1 was based on our finding that predicting vaccine responses solely based on the chronological age

**Table 2. List of Prediction tasks.**

| Task ID | Assay | Task statement | # Models with significant Spearman values | Model with highest Spearman rho (Submission ID) | Top Contributors of winning model |
|---|---|---|---|---|---|
| 1.1 | Antibody titer | Rank the individuals by IgG antibody titers against PT that we detect in plasma 14 days post booster vaccinations. | 7 | user54 | IgG PT |
| 1.2 | Antibody titer | Rank the individuals by fold change of IgG antibody titers against PT that we detect in plasma 14 days post booster vaccinations compared to titer values at day 0. | 20 | user49, avey_2017_M33l, and controlModel_BaselineTask | IgG PT, inflammatory response (M33) BTM |
| 2.1 | Cell frequency | Rank the individuals by predicted frequency of monocytes on day 1 post boost after vaccination. | 11 | user49 | Monocytes, TemraCD4 |
| 2.2 | Cell frequency | Rank the individuals by fold change of predicted frequency of monocytes on day 1 post booster vaccination compared to cell frequency values at day 0. | 2 | user51 | IgG1_PT, CCL3, IL-6 (cytokines), TemCD4, TcmCD8, Monocytes |
| 3.1 | Gene expression | Rank the individuals by predicted gene expression of *CCL3* on day 3 post-booster vaccination. | 11 | user48_1 | MOFA Factor 3 (*CCL3, CXCL8, IL1-B, CCL4*) |
| 3.2 | Gene expression | Rank the individuals by fold change of predicted gene expression of *CCL3* on day 3 post booster vaccination compared to gene expression values at day 0. | 0 | NA | NA |

The tasks are grouped into three main types based on experimental assays: antibody titer tasks, cell frequency tasks, and gene expression tasks. For each group, we asked to rank individual subjects based on either the absolute values of the biological readouts post-vaccination or the fold change compared to pre-vaccination measurement. Task 1.2 demonstrated the highest counts of models (n=20) with significant Spearman correlation coefficients, whereas tasks 2.2 and 3.2 were challenging to predict, with only 2 and 0 models, respectively, showing significant Spearman correlation coefficients.

of the subject (the older, the worse) outperformed a lot of other models in predicting the antibody response to the TdaP vaccination [10]. Therefore, we implemented Control Model 1 simply by ranking subjects on their calendar age. Similarly, Control Model 2 simply ranks participants by the pre-vaccination levels of assay readouts, which we had shown to be highly correlated with post-vaccination levels of the same readouts [14,15]. We implemented this for tasks 1.1 and 1.2, by using the baseline IgG antibody titer values against pertussis toxin as the predictor. For tasks 2.1 and 2.2, we used pre-vaccination monocyte frequencies, and for tasks 3.1 and 3.2, we used pre-vaccine levels of *CCL3* gene expression values. These control models were intended to set a baseline that more complex prediction models should exceed.

Additionally, we implemented a set of 22 literature-derived models developed within the systems vaccinology field that aim to predict vaccination outcomes, as described in Shinde et al [10]. It is important to note that these models were repurposed for our prediction tasks and not evaluated in their original intended areas or studies. Instead, we evaluated these adapted models for their prediction performance on TdaP booster vaccination to determine the generalizability of these predictors. All of the literature models we identified were developed to predict antibody measurements, so we only ran them on Tasks 1.1 and 1.2.

## 4. Contestants' methods to predict vaccine response

We received a total of 25 submissions with the majority (19/25) of teams attempting all six tasks. Two teams completed five tasks, one team completed four tasks, two teams completed two tasks, and one team completed only one task. For simplicity, we refer to all of the computational, mathematical or statistical method used by the submitters to arrive at their ranking as a 'model' - which notably is not meant to imply that these were 'mechanistic models' where different factors have a biological interpretation.

Contestants were asked to describe the methodologies they utilized, which included linear regression, nonlinear regression (regression trees), sparse linear regression, PLS (partial least-squares) or PC (principal component) regression, ensemble/model selection, etc. All methods are listed in Table 3 with a short description that covers data pre- and postprocessing and expanded team summaries can be found in S1 Note. We summarized the submitted models submitted by contestants, categorizing the 25 team submissions into three methodological groups: sparse linear regression, nonlinear regression (e.g., regression trees), and other approaches (e.g., neural networks, AutoML) (Table 3). Sparse linear regression models were the most common, with methods like ElasticNet and LASSO regression frequently paired with multi-omics integration techniques such as MCIA, JIVE and MOFA. For example, model *user48_1* utilized MOFA-based multi-omics integration with handpicked features and LASSO regression. Nonlinear regression approaches included techniques such as categorical boosting, random forests, and block forest regressions. For instance, model *user_54* employed a categorical boosting regression model trained on combined 2020 and 2021 cohorts. Additionally, we included models that did not fall into these two primary categories under "other methods", such as neural networks and automated machine learning pipelines. For example, the model *user34* first identified the most predictive assay for each task and then applied an AutoML model along with clinical data.

Most teams built their models using the provided preprocessed data. Some teams performed additional data processing required as a prerequisite for specific algorithms. These preprocessing techniques included data transformation and scaling (e.g., log10, square root), encoding for categorical features such as race and biological sex (e.g., label, one-hot), data imputation (e.g., PCA, Bayesian), and data normalization.

**Table 3. Summary of methods used in the CMI-PB invited prediction challenge and total points earned.**

| Team | Model ID | Synopsis | Total points earned |
|---|---|---|---|
| **Sparse linear regression** | | | |
| 1 | user_52 | ImputePCA and training and prediction were made using elastic net regression | 3 |
| 2 | user9_3 | Features were reconstructed using JIVE multi-omics integration. Training data consisted of 2020 + 2021 datasets. All four assays and subject information were used, and training and prediction were done using ElasticNet | 1 |
| 3 | user9_4 | Features were reconstructed using JIVE multi-omics integration. Training data consists of 2020 + 2021 datasets. All four assays and subject information were used, and training and prediction were done using ElasticNet CV | 1 |
| 4 | user47 | SuperLearner Ensemble | 2 |
| 5 | user48_1 | Features were using MOFA multi-omics integration, and final features were handpicked instead of solely relying on LASSO regression, training data consisted of 2021 datasets, All four assays and subject information were used, and training and prediction were done using LASSO regression | 5 |
| 6 | user48_2 | Features were using MOFA multi-omics integration, and final features were handpicked instead of solely relying on LASSO regression, training data consisted of 2020 + 2021 datasets, All four assays and subject information were used, and training and prediction were done using LASSO regression | 3 |
| 7 | user5 | Establishing purpose-built models using Multiple Co-inertia Analysis (MCIA), features consist of multi-omics embeddings using MCIA, baseline values of tasks, and clinical and demographic variables. | 3 |
| 8 | user6 | Ensemble approach using SPEAR-constructed supervised multi-omics factors with demographic data | 1 |
| 9 | user9_2 | Multi-omics Integration with JIVE and Lasso | 1 |
| 10 | user_40 | Different regression models on multi-omics data using features from the baseline (day 0) | 1 |
| 11 | user25 | Semi-manual feature selection learned between the 2020↔2021 datasets, followed by linear regression | 3 |
| 12 | user9_1 | Multi-omics Integration with JIVE and Basic Linear Regression | 1 |
| 13 | user49 | Dimension reduction through Multiple Co-inertia analysis and modeled with Linear mixed effects | 8 |
| 14 | user32 | Semi-manual feature selection followed by dimensionality reduction and residual from baseline prediction | 0 |
| 15 | user50 | Semi-manual feature selection followed by dimensionality reduction and residual from baseline prediction | 0 |
| **Nonlinear regression (regression trees)** | | | |
| 16 | user_38 | Categorical boosting Regression model trained on the 2020 training cohort | 3 |
| 21 | user_53 | Categorical boosting Regression model trained on 2021 training cohort | 2 |
| 18 | user_54 | Categorical boosting Regression model trained on 2020+2021 training cohort | 5 |
| 19 | user45 | Model comparison to determine the best algorithm; Manual feature selection; Random forest regression | 2 |
| 20 | user46 | Block forest regression | 0 |
| 21 | user51 | Random forest classifier to simulate training individuals, XGboost to determine the final ranking | 1 |
| 22 | user55 | DecisionTree and Random Forest Regressor | 2 |
| **Other methods (neural network and AutoML etc.)** | | | |
| 23 | user30 | Fully Connected 2-layer neural network with imputation | 0 |
| 24 | user34 | AutoML is based on the most predictive assay or clinical data (trained on 2020 and tested on 2021) | 2 |
| 25 | user34 | AutoML based on the most predictive assay or clinical data (trained on 2020 and tested on 2021) | 2 |
| **Control models** | | | |
| 26 | | Use the age of the study subject as a predictor | 0 |
| 27 | | Utilize the baseline pre-vaccination state of a task as a predictor | 6 |

The 25 team submissions were categorized according to their underlying methodology into Sparse linear regression, Nonlinear regression (regression trees), and other methods. Additional method characterizations are provided in **S1 Note**.

Preprocessing and feature selection are core components of building a predictor. In this challenge, features in the profiling data sets (P) far outnumber the total samples (N), increasing the risk of overfitting. To address this, teams often reduced the number of features modeled by correlating the features in the profiling dataset to the post-vaccination

response data. A few teams also performed multi-omics integration and PC-based techniques to construct combined meta-features. Other preprocessing steps included principal component analysis, categorical regression, regularized regression (e.g., LASSO, ridge, or elastic nets), and mapping gene expression data to biological pathways or transcriptional modules.

Post-processing also differed in the specific models used for individual tasks. Most teams used summarizing or integrating one prediction model for all six tasks. In this approach, models were re-trained for specific tasks and evaluated separately to achieve better performance for each task. Other teams built entirely separate models for each task. Additionally, teams employed various cross-validation approaches, including leave-one-out, k-fold, 5-fold, and cross-cohort (testing on the 2020 cohort and evaluating on the 2021 cohort, and vice versa). Detailed descriptions of the team methods can be found in Table 1.

## 5. Evaluating task performance on vaccine response predictions

We first evaluated the prediction performance of the control models and models from the literature. As specified in the contest description (**Methods**), Spearman's Rank correlation coefficients were utilized as a metric for the evaluation of the submitted models for each task. The prediction tasks in our first challenge involved predicting the rank of individuals in specific immune response readouts, ranging from high to low, after *B. pertussis* booster vaccination based on their pre-vaccination status. By exclusively analyzing relative rankings, the evaluation ensured robust comparisons through ranked predictions without reliance on raw data values.

For Control Model 1 which was solely based on the age of subjects, we found no significant relationship for any of the six tasks (Fig 2A). In contrast, we observed a significant positive correlation for Control Model 2 between the ranking of post-vaccination responses and their respective baselines for all three tasks: Monocytes on day 1, *CCL3* on day 3, and IgG-PT on day 14 (Fig 2A). This suggests that overall, the booster vaccination does not disrupt the pre-existing ranking of subjects in these readouts. In contrast, a strong negative correlation was noted between the fold change of IgG-PT at day 14 and its baseline. This translates to subjects with low pre-vaccination antibody titers showing the largest fold-change increase in titers post-vaccination. Notably, this is not observed for the other two readouts (*CCL3* gene expression and Monocyte frequency levels), suggesting that it is not just a result of '*regression to the mean*'. Rather, individuals with very low antibody titers seem to benefit the most from a booster vaccination.

Of the 22 literature models tested, only four provided a significant Spearman correlation coefficient, and all of those were for task 1.2 (antibody fold-change). None of the literature models outperformed the 'baseline' Control Model 2 (Fig 2B). Overall, this suggests that the Control Models we implemented provided a good baseline that needs to be exceeded by new models to prove their value.

In terms of contestant-submitted predictions, among the 25 submissions received, 20 demonstrated at least one significant correlation coefficient. These models were considered important, and their performances are discussed subsequently (**Section 6**). In the top 20 models, prevalent techniques for selecting predictor genes included univariate feature ranking, meta-gene construction through multi-omics integration, and literature-based gene selection. The common prediction models employed were random forest and regularized regression methods (LASSO and ridge regression), with the latter being notably used by the top-ranked Team 49 in this sub-challenge.

Furthermore, we discuss top models identified for each task using absolute value of spearman corelation coefficients. For task 1.1 (IgG_PT on day 1), seven models

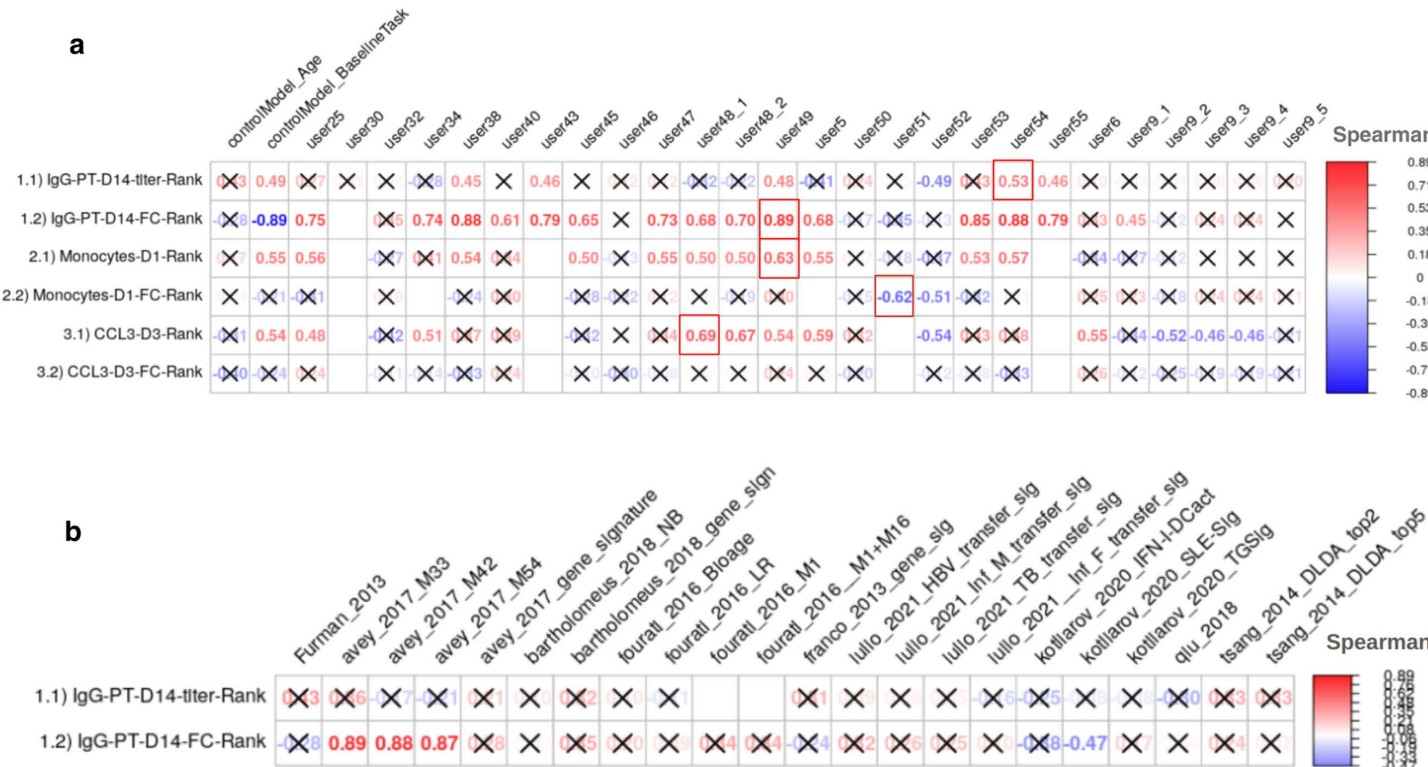

**Fig 2. Evaluation of the prediction models submitted for the invited CMI-PB challenge.** a) Control models and models submitted by contestants b) Models from systems vaccinology literature. Model evaluation was performed using Spearman's rank correlation coefficient between predicted ranks by a contestant and actual rank for each of (1.1 and 1.2) antibody MFI, (2.1 and 2.2) immune cell frequencies, and (3.1 and 3.2) transcriptomics tasks. The number denotes Spearman rank correlation coefficient, while crosses represent any correlations that are not significant using p ≥ 0.05. Red borders around a cell indicate it was the best-performing model for the task.

(*controlModel_BaselineTask, user38, user43, user49, user52, user54, and user55*) demonstrated significant Spearman correlation coefficients, effectively predicting IgG-PT titers on day 14 (Fig 2). These models utilized diverse methodologies. For instance, *user38* and *user54* employed a categorical boosting regression model; *user43* implemented dimension reduction through multiple co-inertia analysis combined with linear mixed-effects modeling; and *user55* incorporated decision tree and random forest regressors (S1 Note). Together, these approaches highlighted the use of both linear and nonlinear methods in this predictive task. The top contributor listed by *user54* was IgG-PT measurements at day 0 from antibody titer assay (Table 2).

Task 1.2 exhibited the highest number of models (n=19) with a significant Spearman correlation coefficient among all evaluated tasks, including the control, literature-based, and contestant-submitted models (Fig 2). The top-performing models for this task were the Control IgG-PT baseline model, the literature model *avey_2017_M33*, and the contestant model submitted by *user49*, all achieving an absolute Spearman coefficient of 0.89. The *avey_2017_M33* model incorporates the inflammatory response blood transcriptional module (M33), featuring genes such as *AIF1*, *APOB48R*, *ARRB2*, *CARD9*, *FCER1G*, *GPX1*, *STAB1*, *TBC1D8*, *TLR4*, and *TNFSF13* (Table 2).

For task 2.1, 11 models demonstrated a significant Spearman correlation coefficient predicting monocyte frequencies on day 1 (Fig 2). These models utilized a variety of approaches, including sparse linear regression methods like LASSO and ElasticNet (by user48), nonlinear

techniques such as categorical boosting (*user38*, *user54*), and ensemble learning methods (*user47*). Additionally, models *user25* and *user49* incorporated dimensionality reduction strategies like MCIA, while *user34* utlised AutoML package (S1 Note). The top predictors listed by model user25 and *user54* were cell frequency assay features such as Monocytes and TemraCD4 (Table 2).

For task 2.2, two models (*user51* and *user52*) demonstrated significant Spearman correlation coefficients, successfully predicting fold change value of monocyte frequencies on day 1 as compared to day 0. Model *user51* utilized a random forest classifier to simulate training individuals and XGBoost for final ranking, while model *user52* applied imputation with PCA followed by training and prediction using elastic net regression (S1 Note). The top predictors listed by model user52 were IgG1_PT, CCL3 (cytokine), IL-6 (cytokine), TemCD4, TcmCD8, Monocytes (Table 2).

For task 3.1, models from 12 models demonstrated significant spearman correlation coefficient by predicting *CCL3* gene expression levels on day 3 (Fig 2). These models employed a range of modeling techniques, including sparse linear regression approaches like ElasticNet and LASSO (by user48_1, user48_2), dimensionality reduction methods (*user49*, *user9*), ensemble learning strategies (*user6*) and *user34* applied AutoML package (S1 Note). The top predictors identified by model *user48_1* included the day 0 value of *CCL3* and a latent factor derived from the MOFA multi-modal model [16]. This latent factor incorporated transcriptomics features such as *CCL3*, *CXCL8*, *IL1B*, and *CCL4* (Table 2). No submissions showed a significant correlation coefficient for task 3.2.

## 6. Top-performing methods include distinct approaches: Multi-omics integration, categorical boosting regression, and subject-based training

Contestant-submitted predictions were aggregated by teams, where the score of each team was calculated using a point system to rank all submissions and identify the overall winner of the challenge. We awarded 3 points to the submission ranked highest in a particular task and 1 point if the contestant attempted the task. The team with the highest total points was awarded as the winner of the challenge. The final scores revealed that the winning team is from the University of Minnesota (Team *user49*), achieving superior predictions in tasks 1.2 (r = 0.7, p-value = 0.001) and 2.1 (r = 0.81, p-value = 0.0031) (Fig 2). Two teams from the La Jolla Institute for Immunology (LJI, Teams *user54* and *user38*) ranked second overall. A team from the National Institutes of Health (NIH, Team *user51*) ranked third overall and achieved the top rank for task 2.2 (see Fig 2A for details). Team *user54* ranked top for task 1.1, and Team 38 ranked top for task 3.1. As no submissions showed a significant correlation coefficient for task 3.2, no team was declared as the winner for that task.

The top-performing team led by Dr. Basu from the University of Minnesota developed a machine learning method that integrated multi-omics profiling data sets and knowledge-enhanced data representations into a probabilistic regression model to learn and predict vaccine response tasks (Fig 3). Starting with raw experimental data, the workflow involved initial data imputation and batch effect correction that considered different time points separately to help maintain the temporal integrity of the data [17]. Feature selection was then performed using various statistical techniques, including LASSO, Ridge, PCA, PLS, and Multiple Co-Inertia Analysis (MCIA). MCIA was then chosen as the best-performing method, integrating different data modalities to produce a reduced set of key multi-omics features [18]. These features were then utilized in a linear mixed-effect model where the MCIA scores were modeled as fixed effects, and the subject-specific effects were captured through a random intercept. The model was trained on a subset of the data, with validation through 5-fold cross-validation to ensure robustness and mitigate overfitting, and then tested to evaluate

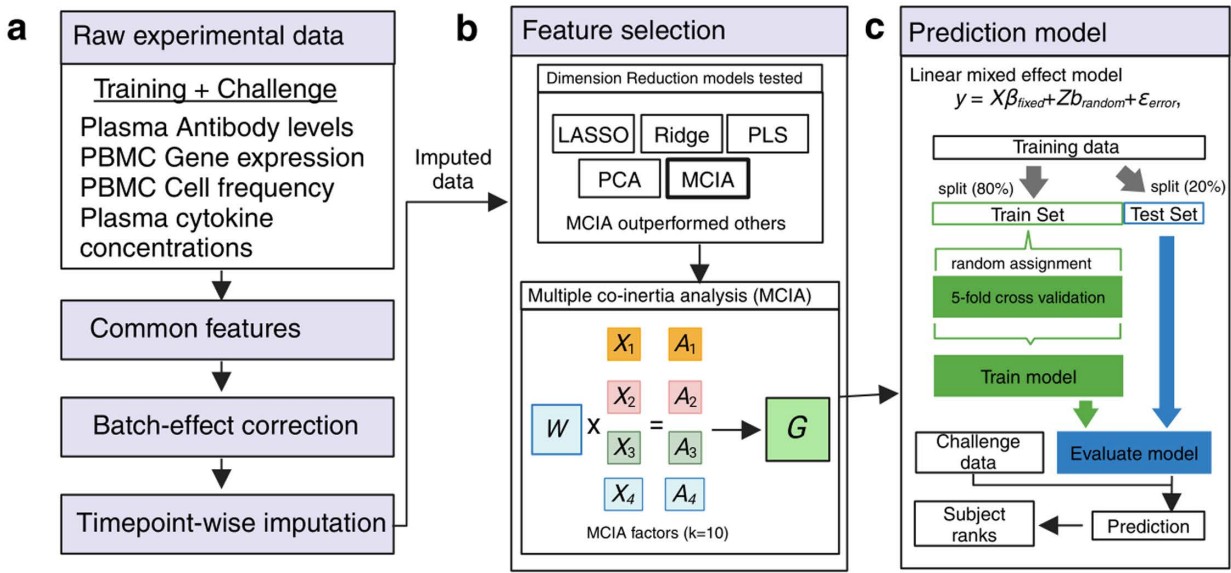

**Fig 3. The method implemented by the winning team.** Schematic overview of the data processing, feature selection, and prediction modeling workflow. (a) The workflow begins with raw experimental data, including training and challenge datasets from plasma antibody levels, PBMC gene expression, PBMC cell frequency, and plasma cytokine concentration assays. The common features across these datasets are identified, followed by batch-effect correction and timepoint-wise imputation. (b) Feature selection was performed using various dimension reduction techniques, including LASSO, Ridge, PLS, PCA, and Multiple Co-inertia Analysis (MCIA). MCIA outperformed the other models and was selected for further analysis. MCIA integrates different data types (e.g., X1, X2, X3, X4) and their associated weights (A1, A2, A3, A4) to produce MCIA factors (G) that represent the combined data structure. (c) These MCIA factors were then used in a Linear Mixed Effects (LME) model to predict the outcome. The model was trained on 80% of the data (train set) using 5-fold cross-validation and evaluated on the remaining 20% (test set). The trained model was then applied to the challenge baseline data to generate predictions, which were used to rank subjects according to their predicted outcomes. Figure is created in https://BioRender.com.

its predictive performance. Predictions were generated using the predict function from the JMbayes package [19], enabling us to forecast outcomes for new subjects with no prior information about their immune response trajectories. This method makes use of empirical Bayes prediction to estimate subject-specific random effects. Overall, Dr. Basu's team developed a purpose-driven machine learning model that integrated multi-omics data and probabilistic regression, employing rigorous preprocessing, feature selection, and validation methods to predict vaccine responses and immune trajectories for new subjects.

There were two second-best-performing teams. The team led by Dr. Thrupp from LJI utilized multi-omics integration with Multi-Omics Factor Analysis (MOFA) which is also a factor analysis model that provides a general framework for the integration of multi-omic data sets in an unsupervised fashion [20]. Initially, processed data from the 2020 and 2021 training cohorts, which included all four assays, were used to construct 10 MOFA factors, which represent condensed, biologically meaningful summaries of the multi-omics data (Fig C in S1 Text). These multi-omics factors allowed to reduce dimensionality while retaining essential information for downstream predictive modeling. Subsequently, LASSO based regularization was employed to identify the best-performing features by penalizing less informative variables, ensuring the model focuses on the most relevant contributors to immune response prediction [21]. The model was trained on a subset of this data, validated through 5-fold cross-validation, and then tested to assess its predictive performance. By integrating MOFA-derived factors with LASSO feature selection and rigorous validation, Dr. Thrupp's team demonstrated a compelling approach to leveraging multi-omics data for predictive modeling.

The team led by Dr. Mahita from LJI utilized the Categorical Boosting (*CatBoost*) Regression model, a machine learning algorithm specifically designed to handle categorical features efficiently. This model was trained on the 2020 and 2021 training cohorts [22] (Fig D in S1 Text). Feature selection was conducted manually, selecting features that exhibited consistent Spearman correlation coefficients when the model was trained separately on the 2020 and 2021 datasets. This approach ensured that only the most reliable and stable features were used for the final model, aiming to enhance the robustness and accuracy of the predictive outcomes. This purpose-driven feature selection process, combined with the powerful *CatBoost* algorithm, allowed Dr. Mahita's team to achieve strong predictive performance while ensuring the generalizability of their model across different datasets.

The third-ranked team led by Dr. Gibson from the NIH adopted a distinctive strategy by employing a Random Forest classifier and *XGBoost* [23,24]. The Random Forest classifier was used to simulate the process of training models on individual subjects, allowing the team to capture subject-specific variations. XGBoost, a powerful gradient boosting framework, was then employed to refine the model and determine the final rankings, leveraging its ability to handle large datasets and complex interactions between multi-omics features. They attempted four of the six tasks, specifically excluding the gene expression tasks. This team utilized processed data from three assays: cell frequency, cytokine concentrations, and antibody measurements (Fig E in S1 Text). For tasks 1.1 and 1.2, they addressed missing values through data imputation using the median of each antibody feature to ensure consistency in the dataset and minimize the impact of missing values on the model's predictive performance. To validate their model, they applied K-fold validation, ensuring the robustness and reliability of their predictive model through systematic resampling and evaluation. By combining Random Forest classifier with *XGBoost* with a robust validation strategy, Dr. Gibson's team demonstrated a well-rounded approach to achieving reliable predictions for ranked immune response outcomes.

## Discussion

In this study, we evaluated multi-omics data from Tdap booster immunizations to predict vaccine outcomes. We focused on *B. Pertussis* because of its continued public health importance and the ability to compare different vaccination regimes. *B. Pertussis* causes whooping cough, a highly contagious respiratory infection that most severely affects infants [25]. The introduction of whole-cell pertussis (wP) vaccines in ~1950 massively reduced the incidence of infections. Due to observed reactogenicity side effects, the wP vaccines were replaced with acellular pertussis (aP) vaccines in 1996. Following this, pertussis incidence has been rising in the last two decades, likely due to waning immunity post aP vaccination [26–30]. Studies, including our own [11,31,32], have shown long-lasting effects and differences in T cell responses in adults originally vaccinated with aP versus wP vaccines, despite subsequent aP booster vaccination, but it remains unclear how these differences are maintained over time [33,34]. To address these questions, our near-term goal is to determine how an individual responds to pertussis antigen re-encounter by characterizing the resulting cascade of events (i.e., recall memory response) and relating it to the pre-vaccination immune state.

This 'invited' challenge differed from our first 'dry run' challenge by including teams from labs other than the organizers. Insights gained from all 49 submitted methods and their relative performance provide a valuable resource for future algorithm development (Table 2, S1 Note). We observed that the top-performing methods employed distinct and innovative approaches to the challenge. These included strategies such as multi-omics data integration, which leverages the combined information from multiple omics to enhance predictive power; categorical gradient boosting regression, which effectively handles discrete outcome variables;

and subject-based training, where models were tailored to individual-specific characteristics to improve accuracy in predicting vaccine responses. The diversity of these successful methodologies highlights the complex and multifaceted nature of TDaP booster vaccination response prediction and emphasizes the importance of adopting various approaches to tackle this challenge effectively.

Furthermore, the presented results-based models showed significant Spearman correlation coefficients. Contestants employed diverse methods that included different composite features through both supervised (e.g., BTMs) and unsupervised (e.g., PCA, MOFA, MCIA) approaches. The diversity in methodology reflects the contestants' attempts to capture the complex and multi-dimensional nature of the data. A critical component in the success of these models was the approach to data preprocessing. Key steps, such as normalization, handling missing values, and feature scaling, were employed by most models (19/25) to ensure the data was adequately prepared for analysis. These preprocessing techniques are known to reduce biases, standardize the data, and optimize it for model training [35]. Overall, effective data preprocessing played a crucial role in the improved performance and reliability of the predictive models.

We observed that the control models we established, which relied on the subject's age and pre-vaccination state as task variables, performed well as baseline models for comparing more complex models submitted by contestants. Modeling post-vaccine immune responses involves significant variability due to individual differences in immune system behavior, the influence of prior exposures, and other unknown confounding factors [15,36,37]. Despite these complexities, it was essential to construct robust baseline models that captured the fundamental biological responses using minimal variables. By focusing on straightforward, readily available variables such as age and pre-vaccination state, we were able to create a reliable reference point. This allowed us to accurately assess how more complex models, incorporating immunological and demographical data, predicted post-vaccine responses. These baseline models thus played an important role in evaluating the complexity of the advanced approaches while providing a fair comparison.

The IgG-PT-D14 tasks (both value and fold change) received the highest number of models with significant correlations, indicating that these tasks were the most successfully predicted. One possible reason for this success could be that antibody responses have been well-characterized immune response biomarkers following vaccination [14,38]. Many studies have well-established antibody response as a reliable correlate of protection against many pathogens, and immunological signatures, including transcriptional signatures, have been reported to predict antibody responses to several vaccines, including yellow fever and influenza [7,14]. For instance, studies on influenza vaccination have found that elevated levels of IgG and certain cytokines like IFN-γ and IL-6 correlate with better protection against the virus [39–41]. The robust performance of models predicting IgG responses in our study aligns with these prior studies, reinforcing the predictive power of top antibody-based model in vaccine response assessment.

In contrast, the Monocyte-D1 and *CCL3*-D3 (tasks 2.1 and 3.1) response prediction tasks exhibited a mix of results, with some models performing well while others demonstrated inconsistent performance, indicating moderate difficulty in predicting these tasks. Additionally, the Monocyte-D1 and *CCL3*-D3 (tasks 2.2 and 3.2) fold change tasks had fewer models showing significant correlations, suggesting greater difficulty or variability in their prediction. The inconsistent performance could be due to the complex nature of monocyte and gene expression responses, which are influenced by many factors such as cellular interactions, signaling pathways, and individual immune system nuances. Fold-change calculations may amplify this complexity because they are sensitive to baseline levels; small errors or fluctuations at the baseline can lead to significant discrepancies in the fold-change outcome. These

mixed results underscore the need for innovative modeling techniques to better capture the nuances of monocyte and *CCL3* responses, specifically fold-change values. Overall, the variability in prediction success across underlying tasks highlights the inherent challenge of modeling TdaP post-vaccination immune responses, particularly when compared to the more predictable IgG responses. This underscores the need of advancing our understanding and modeling capabilities to address the complexities associated with cellular immune response predictions in future vaccine studies.

We created a hub for researchers to push for novel models of immunity for predicting outcomes of TdaP boost. We expect the resultant models will also be relevant for other vaccinology studies. Researchers who are interested in participating in the third challenge are encouraged to check the website (www.cmi-pb.org) for the upcoming contest information. The findings of these models can be used as a guide to advance vaccine development by providing a broader understanding of the immune system and identifying biomarkers and pathways that can be targeted with an optimized vaccination strategy. These biomarkers and pathways could be targeted by altering the vaccine formulation, e.g., the use of vaccine adjuvants. Additionally, computational models could predict which cell populations respond well to which vaccines and why, thereby providing knowledge that could be used for a more personalized vaccination strategy. To conclude, computational models can provide novel insights that ultimately lead to more effective and durable vaccines.

## Methods

### Ethics statement

This study was performed with approvals from the Institutional Review Board at the La Jolla Institute for Immunology, and written informed consent was obtained from all participants before enrollment (protocol number VD-101).

### Challenge data and ground truth

The invited CMI-PB prediction challenge is outlined in Fig 1. A total of three multi-omics datasets were provided to contestants consisting of 117 subjects. The entire dataset was split into training and challenge datasets. The training dataset includes two independent cohorts, the 2020 cohort and the 2021 cohort, and these cohorts are discussed in detail in two recent publications: da Silva Antunes et al. [11] and Shinde et al. [10], respectively. The challenge or ground truth evaluation dataset consists of 21 subjects, and we conducted experimental assays similar to those performed on the training datasets, as described in the following:

**Experimental model and subject details.** The characteristics of all 21 subjects are summarized in Table A in S1 Text, with human volunteers who had received either the aP or wP vaccination during childhood being recruited for the study. All participants provided written informed consent before donation and were eligible for TdaP (aP) booster vaccination. Longitudinal blood samples were collected pre-booster vaccination (day -30, -14, 0) and post-booster after 1, 3, 7, and 14 days.

**Experimental data generation.** Each multi-omics dataset consists of metadata about subjects and experimental data generated using four assays: plasma antibody measurements, PBMC cell frequencies, plasma cytokine concentrations, and RNA sequencing. We run experiments on three pre-booster (day -30, -14, 0) timepoints and four post-vaccine responses (day 1, 3, 7, and 14) time points.

1) **Plasma antibody measurements.** An indirect serological assay was employed using xMAP Microspheres (Luminex Corporation) to measure TdaP antigen-specific antibody

responses in human plasma. Pertussis antigens (PT, PRN, Fim2/3, FHA), Tetanus Toxoid (TT), Diphtheria Toxoid (DT), and Ovalbumin (negative control) were coupled to uniquely coded beads (xMAP MagPlex Microspheres). Antibody details are shown in Table B in S1 Text. A detailed description is provided by da Silva Antunes et al. [11].

2) **PBMC cell frequencies.** Twenty-one different PBMC cell subsets were identified using manual gating using FlowJo (BD, version 10.7.0). The detailed description is provided by da Silva Antunes et al. [11] and depicted as Fig F in S1 Text.

3) **Plasma cytokine concentrations.** Plasma samples were randomly distributed on 96 well plates for quantification of different plasma cytokines by Olink proteomics assay. The detailed description is provided by da Silva Antunes et al. [11].

4) **RNA sequencing.** Library preparation was performed using the TruSeq Stranded mRNA Library Prep Kit (Illumina). Libraries were sequenced on a HiSeq3000 (Illumina) system. The paired-end reads that passed Illumina filters were further filtered for reads aligning to tRNA, rRNA, adapter sequences, and spike-in controls. The remaining reads were aligned to the GRCh38 reference genome and Gencode v27 annotations using STAR (v2.6.1) [42]. DUST scores were calculated with PRINSEQ Lite (v0.20.3) [43], and low-complexity reads (DUST >4) were removed from the BAM files. The alignment results were parsed via the SAMtools to generate SAM files [44]. Read counts to each genomic feature were obtained with the featureCounts (v1.6.5 using the default options along with a minimum quality cut-off (Phred >10)) [45].

Contestants were supplied with the baseline immunoprofiling data for all challenge dataset subjects. The post-vaccine response data, which contain the ground truth, were hidden from the contestants.

## Data processing

In addition to the original raw data generated by immunoprofiling, we performed data pre-processing as described in Fig 1. In addition to the original raw data generated by immunoprofiling, we performed data pre-processing as described in Fig 1. First, we identified common features between the training and challenge datasets and excluded features with a coefficient of variance less than 0.3. Second, we performed baseline normalization on cell frequency, antibody measurement, and cytokine concentration data. Specifically, all zero values were replaced with the limit of detection (LOD) to account for background noise where the LOD was calculated as half the value of the first non-zero measurement for each feature. The median baseline concentration was first calculated using the adjusted dataset. This value was then used to normalize all data points by dividing each value by the median baseline concentration. Third, we ran ComBat-seq with default parameters to correct batch effects [12]. To maintain consistency, we performed baseline normalization on cell frequency, antibody measurement, and cytokine concentration data in the test dataset but did not apply any normalization to the gene expression data.

**Formulating the prediction tasks.** Contestants were challenged to predict a ranked list of the highest response (to be ranked first) to the lowest response (to be ranked last) subjects for each prediction task provided. We formulated six prediction tasks in order to quantitatively compare different approaches to model immune responses to TdaP booster vaccination. We selected biological readouts known to be altered by booster vaccination under the premise that these readouts would likely capture meaningful heterogeneity across study subjects based on our previous work [11]. We formulated six prediction tasks: three required contestants to predict specific biological readouts on particular days following the vaccine response, and

the other three required contestants to predict the fold change between specific biological readouts on particular days following the vaccine response and the pre-vaccination state.

Tasks 1.1 and 1.2 (IgG levels against PT antigen on Day 14) were chosen for plasma IgG levels, as they typically mark the peak antibody response period around two weeks post-booster vaccination [11]. In task 1.1, contestants had to predict plasma IgG levels against PT on day 14 post-booster vaccination. For task 1.2, contestants were required to predict the fold change of the plasma IgG levels against PT between day 14 post-booster vaccination and baseline. Further, we have shown that the percentage of monocytes was significantly elevated on day 1 post-booster vaccination compared to baseline (i.e., before booster vaccination), highlighting the role of monocytes in Tdap vaccine response [11]. Tasks 2.1 and 2.2 required contestants to predict the overall frequency of monocytes among PBMCs on day 1 post-booster vaccination and the corresponding fold change, respectively. Similarly, our previous finding was that a subset of aP-primed individuals showed an increased expression of proinflammatory genes, including *CCL3*, on day 3 post-booster vaccination [11]. Tasks 3.1 and 3.2 required contestants to predict *CCL3* gene expression on day 3 post-booster vaccination and the corresponding fold change values compared to baseline.

**Prediction challenge evaluation.** After receiving the contestants' ranked predictions, we curated the rank file. If we found NA values in the ranked list, we imputed them with the median rank for that list. Evaluations were performed in two steps.

First, we chose the Spearman rank correlation coefficient as evaluation metric to compare the predicted ranked list ($p$) for each task, $t$, and n subjects (n=21 for the set of challenge dataset subjects), $R\_(p,t) = (r\_(p,1), r\_(p,2), ..., r\_(p,n))$ against ground truth ($g$) ranked list R_(g,t) = (r_(g,1), r_(g,2), ..., r_(g,n)). The Spearman rank correlation coefficient ($\rho$) is given by:

$$\rho_{(R_{(g,t)}, R_{(p,t)})} = 1 - \left( 6\sum(i=1)^n d_i^2 \right) / \left( n(n^2 - 1) \right)$$

where d_i = R_(g,i) - R_(p,i) is the difference between the ranks of each pair. In this way, each task submitted by constant was evaluated. We used the Spearman correlation coefficient as the evaluation metric, as it is non-parametric and well-suited for data with non-normal distributions, which were characteristic of the analyte measurement outputs from the different experimental assays utilized in the study.

Second, we devised a point system to rank all submissions and identify the overall winner of the challenge. Specifically, we awarded 3 points if a submission was top-ranked in a particular task and 1 point if the contestant attempted the task. This point system was used to identify winners of the challenge.

**Quantification and statistical analysis.** Statistical analyses are detailed for each specific technique in the specific Methods section or in the figure legends, where each specific comparison is presented. Statistical tests were performed using R (version 4.1, www.r-project.org/) of the Spearman correlation coefficient. Details pertaining to significance are also noted in Fig 2 legends, and $p < 0.05$ is defined as statistical significance.

## Limitations of the study

Our challenge dataset cohort comprised multi-omics data for 21 subjects, a size smaller than the training data cohorts of 96 subjects, maintaining just an 80:20 training-to-challenge dataset ratio. The smaller size of the challenge cohort may result in reduced precision and heightened sampling variation in Spearman rank calculations, which were used as an evaluation matrix, potentially impacting the reliability and generalizability of correlation results.

However, models developed by contestants exhibited strong performance, surpassing control models in four tasks. To address this limitation, we plan to expand size of the challenge cohort and include additional assays related to T-cell responses [46] in future contests. A larger cohort will reduce sampling variation, increase statistical power, and improve model generalizability by better representing biological variability [47]. The inclusion of additional assays will further enrich the dataset, providing a more comprehensive view of immune response variability.

Another limitation of this study was that the participants were invited based on a selection process that was informed by the investigators leading this study, which will likely have missed relevant additional participants that would have conducted different modeling approaches. This will be addressed in the follow up challenge, which will be open to all participants.

## Supporting information

**S1 Text. Summary of feedback received after the conclusion of the challenge. Table A**. Antibody information. **Table B**. The characteristics of all 21 subjects in the challenge dataset. **Fig A: Plot of assay data before and after normalization and batch effect correction**. For each assay, the plots on the left represent data before batch correction, while the plots on the right represent data after normalization and batch correction. **Fig B: Schematic representation of control model construction.** (a) Control model using subject age: Raw age values (orange) are ranked (blue) to create a ranked list based on age. (b) The control model uses IgG-PT levels on Day 0 for Task 1.1: Raw IgG-PT values (red) are ranked (blue) to represent the relative position of subjects based on their pre-vaccination IgG-PT levels. Similarly, monocyte frequency on Day 1 was used to construct the control model for Task 2.1, and *CCL3* levels on Day 3 were used to construct the control model for Task 3.1. (c) Control model using the inverse of IgG-PT levels on Day 0 for Task 1.2: Raw IgG-PT values (red) are converted into negative values, then ranked (blue), emphasizing individuals with lower IgG-PT levels. Similarly, the negative values of monocyte frequency on Day 1 were used to construct the control model for Task 2.2, and the negative values of *CCL3* levels on Day 3 were used to construct the control model for Task 3.2. **Fig C: The method implemented by the Dr. Thrupp's team (user48_1).** (a) The analysis pipeline begins with batch-corrected data for both training and prediction phases. The training data includes all four assay data provided, i.e., Plasma Antibody levels, PBMC Gene expression, PBMC Cell frequency, and Plasma Cytokine concentrations. Similarly, prediction data is derived from the same types of measurements. b) Multi-omics factor Analysis (MOFA) is employed for multi-omics data integration through dimensionality reduction. The input data (X1, X2, X3, X4) represent different omics datasets (e.g., gene expression, cell frequency, cytokine levels, and antibody levels), each associated with a corresponding matrix (A1, A2, A3, A4). MOFA outputs a set of factors (G), which are used for subsequent prediction modeling. (c) Lasso regression is used to predict the output using the selected features. The dataset is split into training (70%) and test (30%) sets. The training data undergoes leave-one-out cross-validation (LOOCV) for model training, followed by prediction on the test set. The model's performance is evaluated on the challenge baseline data, and subjects are ranked based on the challenge data outcomes submitted on the submission portal for evaluation. **Fig D: The method implemented by Dr. Mahita's team (user54).** Step 1: Data preprocessing and exploration were performed by combining subject and specimen metadata and selecting specific assays. Step 2: Leveraging longitudinal data by calculating fold change between key time points for each task. Step 3: Categorical Boosting (CatBoost) algorithm was chosen for predicting fold change or differences, with Spearman's correlation used for model evaluation. Step 4: Feature selection through model training and testing across

cohort-specific datasets, with consistent correlation coefficients guiding the final feature combination. **Fig E: The method implemented by Dr. Gibson's team (user51).** The XGBoost algorithm workflow was developed to rank individual input subjects based on Day 14 IgG PT levels. The analysis includes 20 features from cell frequency assay, covering Monocytes (f = 4), T cell subsets (f = 12), B cells (f = 1), and Innate immune cells (f = 3); 30 Olink cytokine features such as CCL4, IL-18, and CXCL11; and 32 antibody features, including total IgG, IgG1-4, and the sum of IgG for PT, TT, PRN, FHA, DT, and OVA. (a) Input subjects (e.g., Subject 1 and Subject 90) are evaluated using a decision tree model. Each subject is assigned a classification, such as "Ranked Lower" or "Ranked Higher," based on their IgG PT level. The classification output is combined with a probability score (e.g., Subject 1: Prediction = -1, Probability = 0.9; Subject 90: Prediction = 1, Probability = 0.7), and the final prediction is calculated as the product of the prediction and probability. (b) The XGBoost algorithm processes a dataset of model summaries shown in Panel C through a series of decision trees. The algorithm iteratively refines predictions by calculating residuals after each tree and performing node splitting based on an objective function, improving accuracy over multiple trees (Tree1, Tree2,..., Treek). The final rank prediction for each subject is determined by summing the outputs of all trees. (c) A table displays the predicted values for various subjects (subject_id) across different models (e.g., model_1, model_3), highlighting the individualized outcomes of the prediction process. **Fig F**. Gating strategy for PBMC cell frequencies (FACS).
(PDF)

**S1 Note. A detailed description of CMI-PB invited prediction challenge methods.**
(XLSX)

## Acknowledgments

We are grateful to the La Jolla Institute for Immunology Flow Cytometry and Bioinformatics core facilities for their services. The authors would like to thank all donors who participated in the study and the clinical studies group staff - particularly Gina Levi - for all the invaluable help.

## Author contributions

**Conceptualization:** Bjoern Peters.

**Data curation:** Pramod Shinde, Lisa Willemsen, Minori Aoki, Jiyeun Lee.

**Formal analysis:** Pramod Shinde, Bjoern Peters.

**Funding acquisition:** Bjoern Peters.

**Investigation:** Pramod Shinde, Lisa Willemsen, Bjoern Peters.

**Methodology:** Pramod Shinde, Lisa Willemsen, Minori Aoki, Jiyeun Lee, Somayeh Nili, Shelby Orfield, James A. Overton, Bjoern Peters.

**Resources:** Pramod Shinde, Jason A Greenbaum, Barry Grant.

**Supervision:** Leying Guan, Ferhat Ay, Barry Grant, Steven H Kleinstein, Bjoern Peters.

**Writing – original draft:** Pramod Shinde, Lisa Willemsen, Saonli Basu, William Gibson, Mari Kojima, Jarjapu Mahita, Shelby Orfield, Nicola Thrupp, Bjoern Peters.

**Writing – review & editing:** Pramod Shinde, Lisa Willemsen, Michael Anderson, Minori Aoki, Saonli Basu, Julie G Burel, Peng Cheng, Souradipto Ghosh Dastidar, Aidan Dunleavy, Tal Einav, Jamie Forschmiedt, Slim Fourati, Javier Garcia, William Gibson, Jason A Greenbaum, Leying Guan, Weikang Guan, Jeremy P Gygi, Brendan Ha, Joe Hou,

Jason Hsiao, Yunda Huang, Rick Jansen, Bhargob Kakoty, Zhiyu Kang, James J Kobie, Mari Kojima, Anna Konstorum, Jiyeun Lee, Sloan A Lewis, Aixin Li, Eric F Lock, Jarjapu Mahita, Marcus Mendes, Hailong Meng, Aidan Neher, Somayeh Nili, Lars Rønn Olsen, Shelby Orfield, James A. Overton, Nidhi Pai, Cokie Parker, Brian Qian, Mikkel Rasmussen, Joaquin Reyna, Eve Richardson, Sandra Safo, Josey Sorenson, Aparna Srinivasan, Nicola Thrupp, Rashmi Tippalagama, Raphael Trevizani, Steffen Ventz, Jiuzhou Wang, Cheng-Chang Wu, Ferhat Ay, Barry Grant, Steven H Kleinstein, Bjoern Peters.

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
