## [Decision Letter · Decision Letter 0]

4 Dec 2024

PCOMPBIOL-D-24-01620

Putting computational models of immunity to the test - an invited challenge to predict B.pertussis vaccination responses

PLOS Computational Biology

Dear Dr. Shinde,

Thank you for submitting your manuscript to PLOS Computational Biology. After careful consideration, we feel that it has merit but does not fully meet PLOS Computational Biology's publication criteria as it currently stands. Therefore, we invite you to submit a revised version of the manuscript that addresses the points raised during the review process.

Please submit your revised manuscript within 30 days Feb 03 2025 11:59PM. If you will need more time than this to complete your revisions, please reply to this message or contact the journal office at ploscompbiol@plos.org. Please include the following items when submitting your revised manuscript:

We look forward to receiving your revised manuscript.

Kind regards,

Jessica M. Conway

Academic Editor

PLOS Computational Biology

Rob De Boer

Section Editor

PLOS Computational Biology

Feilim Mac Gabhann

Editor-in-Chief

PLOS Computational Biology

Jason Papin

Editor-in-Chief

PLOS Computational Biology

**Journal Requirements:**

At this stage, the following Authors/Authors require contributions: Pramod Shinde, Lisa Willemsen, Michael Anderson, Minori Aoki, Saonli Basu, Julie G Burel, Peng Cheng, Souradipto Ghosh Dastidar, Aidan Dunleavy, Tal Einav, Jamie Forschmiedt, Slim Fourati, Javier Garcia, William Gibson, Jason A Greenbaum, Leying Guan, Weikang Guan, Jeremy P Gygi, Brendan Ha, Joe Hou, Jason Hsiao, Yunda Huang, Rick Jansen, Bhargob Kakoty, Zhiyu Kang, James J Kobie, Mari Kojima, Anna Konstorum, Jiyeun Lee, Sloan A Lewis, Aixin Li, Eric F Lock, Jarjapu Mahita, Marcus Mendes, Hailong Meng, Aidan Neher, Somayeh Nili, Lars Rønn Olsen, Shelby Orfield, James Overton, Nidhi Pai, Cokie Parker, Brian Qian, Mikkel Rasmussen, Joaquin Reyna, Eve Richardson, Sandra Safo, Josey Sorenson, Aparna Srinivasan, Nicola Thrupp, Rashmi Tippalagama, Raphael Trevizani, Steffen Ventz, Jiuzhou Wang, Cheng-Chang Wu, Ferhat Ay, Barry Grant, Steven H Kleinstein, and Bjoern Peters. Please ensure that the full contributions of each author are acknowledged in the "Add/Edit/Remove Authors" section of our submission form.

Potential Copyright Issues:

i) Figure 1. Please confirm whether you drew the images / clip-art within the figure panels by hand. If you did not draw the images, please provide (a) a link to the source of the images or icons and their license / terms of use; or (b) written permission from the copyright holder to publish the images or icons under our CC BY 4.0 license. Alternatively, you may replace the images with open source alternatives. See these open source resources you may use to replace images / clip-art:

6) Thank you for providing us with your Data Availability Statement. We found that this link "https://www.cmi-pb.org/dataandathttps://www.cmi-pb.org/downloads/cmipb_challenge_datasets/legacy/2nd_challenge/2024-02-02/2nd_challenge/"  reaches a 404 error page. Please amend it to a new link or provide further details to locate the data.

7) Please amend your detailed Financial Disclosure statement. This is published with the article. It must therefore be completed in full sentences and contain the exact wording you wish to be published.

1) State what role the funders took in the study. If the funders had no role in your study, please state: "The funders had no role in study design, data collection and analysis, decision to publish, or preparation of the manuscript.".

If you did not receive any funding for this study, please simply state: The authors received no specific funding for this work.

**Reviewers' comments:**

Reviewer's Responses to Questions

Reviewer #1: In this article, the authors established a community resource to challenge computational models that predict the outcome of booster B.pertussis vaccination from a select group of scientists, evaluate and compare those models in their performance in predicting B. pertussis booster responses. They looked for the features of successful models and identified objects based on the level of difficulty in being predicted. The goal, method and results are well stated and clear. There are a few things in the paper that need to be clarified, for this reason I would suggest this paper to undertake some minor revision.

1. In this ‘invited challenge’, what is the criteria in selecting the scientists and their modeling work?

2. Are all main types of models of predicting immune response covered in this study?

3. The tasks to be evaluated consisted of predicting plasma IgG levels against the pertussis toxin (PT) on day 14 post-booster vaccination and the fold change of the plasma IgG levels against PT between day 14 post-booster vaccination and baseline, predicting the overall frequency of monocytes among PBMCs on day 1 post-booster vaccination and the corresponding fold change and the CCL3 gene expression on day 3 post-booster vaccination and the corresponding fold change values compared to baseline. Is there a reason that those prediction are tested at those specific days (14, 3, 1 post booster vaccine)?

4. Are the evaluation and comparison conducted on the list of ranks for the 21 subjects for each one of the 6 tasks? Is there any evaluation on the values of plasma IgG levels, overall frequency of monocytes among PBMCs or the CCL3 gene expression?

Reviewer #2: Shinde et al. provide a nice overview of the third competition in the Computational Models of Immunity to Pertussis Booster (CMI-PB) challenge series. In their paper, the authors discuss the outcomes of a model development/prediction task undertaken by 20 teams, that also included two "control" models and 22 literature based models. The goal of this competition is noteworthy and raises important questions about the applicability of models developed to understand pertussis vaccination. I have some suggestions for additions that I encourage the authors to consider:

1. Clarifying what "model" means here would be useful for a broader audience. The winning team used machine learning, regression models. As a mechanistic modeller, I was a bit unsure throughout the text what the structure of the models were and were they statistical vs mathematical vs computational etc. The authors do provide a list of the top 5 + "other" model types, but some more discussion would help clarify.

2. Similarly, the control models were unclear to me. Control model 1 used age to rank (?) participants it seems. Is it a regression model with age as the regressor? Maybe a figure panel or equations laying out both control models would be useful.

3. I was interested in the commonalities between the highest ranked models. Were there any biological features of well-ranked (say top 5) conserved across models. If so, what do they help us learn about pertussis booster vaccination. Some overview is given in the discussion. However, given the unique nature of this challenge and the data the team has assembled, it would be nice to also provide key biological take-away messages.

4. For the normalization, was there any removal of background measures when calculating the median baseline concentration?

I also had a few minor comments:

1. On page 4, it may be clearer to move the sentence "The repeat pre-vaccination samples were intended to give a stable estimate of baseline and variability." to after the discussion of the measurement taken from each specimen (otherwise one wonders "baseline and variability of what?").

2. Also on page 4, "such as IL-6 cytokine in the plasma cytokine" could simply be "such as IL-6 in the plasma..."

3. I think there is a missing reference on page 5 ("[38490204]").

4. Page 11: "The detailed description is provided here11." Maybe "is provided in da Silva Antunes et al.11" is better?

Reviewer #3: The manuscript presents an important contribution to the field of systems vaccinology. It effectively demonstrates the value of community-driven benchmarking in computational modeling of vaccine responses and provides a robust framework for evaluating predictive models. The study is well-designed, and its findings offer meaningful insights into the methodologies that improve predictive accuracy and reliability in this context.

The manuscript is well-written, and the dataset and methodology are clearly described. The open science approach, with datasets and code made publicly available, aligns with best practices for transparency and reproducibility in research. The establishment of a community resource and the invitation for future challenges further enhance the relevance and impact of the study.

There are only a few minor points that could be addressed to strengthen the clarity and accessibility of the manuscript:

1. Tables are somewhat dry (i.e. “boring”) in presentation. It would be helpful to visualize some of these results for better engagement and understanding.

2. The manuscript acknowledges the limited size of the challenge dataset, but including a brief discussion on how this limitation could be addressed in future challenges would provide a more balanced perspective. Additionally, it would be beneficial to comment on the dataset specificity of the models and their potential generalizability.

3. Adding a few sentences to highlight how the findings, particularly from a biological perspective, align with or diverge from previous systems vaccinology studies would enhance the contextual significance of the work.

4. While the conclusion is robust, it could be strengthened by outlining how the findings might influence vaccine development or immunological research.

**Have the authors made all data and (if applicable) computational code underlying the findings in their manuscript fully available?**

Reviewer #1: Yes

Reviewer #2: Yes

Reviewer #3: Yes

PLOS authors have the option to publish the peer review history of their article (what does this mean? ). If published, this will include your full peer review and any attached files.

**Do you want your identity to be public for this peer review?** For information about this choice, including consent withdrawal, please see our Privacy Policy .

Reviewer #1: No

Reviewer #2: No

Reviewer #3: No

**Figure resubmission:**
---

## [Decision Letter · Decision Letter 1]

4 Mar 2025

Dear Dr. Shinde,

We are pleased to inform you that your manuscript 'Putting computational models of immunity to the test - an invited challenge to predict B.pertussis vaccination responses' has been provisionally accepted for publication in PLOS Computational Biology.

Before your manuscript can be formally accepted you will need to complete some formatting changes, which you will receive in a follow up email. A member of our team will be in touch with a set of requests. Please also correct the typos flagged by Reviewer 2.

Best regards,

Jessica M. Conway

Academic Editor

PLOS Computational Biology

Rob De Boer

Section Editor

PLOS Computational Biology

Please correct the typos flagged by Reviewer 2.

Reviewer's Responses to Questions

**Comments to the Authors:**

Reviewer #1: Thank you for your revisions. The concerns raised in my previous review have been adequately addressed, and I have no further comments.

Reviewer #2: The authors have responded to all of my questions and I recommend acceptance.

Just a couple typos that I noticed in the revised manuscript:

-page 9: "For task 2.2, models two models"

-page 15: "Tasks 1.1 and 1.2 (IgG levles)"

**Have the authors made all data and (if applicable) computational code underlying the findings in their manuscript fully available?**

Reviewer #1: None

Reviewer #2: None

PLOS authors have the option to publish the peer review history of their article (what does this mean? ). If published, this will include your full peer review and any attached files.

**Do you want your identity to be public for this peer review?** For information about this choice, including consent withdrawal, please see our Privacy Policy .

Reviewer #1: No

Reviewer #2: No

---

## [Editor Report · Acceptance letter]

PCOMPBIOL-D-24-01620R1

Putting computational models of immunity to the test - an invited challenge to predict B.pertussis vaccination responses

Dear Dr Shinde,

I am pleased to inform you that your manuscript has been formally accepted for publication in PLOS Computational Biology. Your manuscript is now with our production department and you will be notified of the publication date in due course.

With kind regards,

Zsofia Freund
